# Effect of a Multi-Ingredient *Coriolus-versicolor*-Based Vaginal Gel in Women with HPV–Dependent Cervical Lesions: The Papilobs Real-Life Prospective Study

**DOI:** 10.3390/cancers15153863

**Published:** 2023-07-29

**Authors:** Javier Cortés Bordoy, Javier de Santiago García, Marta Agenjo González, Damián Dexeus Carter, Gabriel Fiol Ruiz, Carmen García Ferreiro, Silvia P. González Rodríguez, Marta Gurrea Soteras, Ester Martínez Lamela, Santiago Palacios Gil-Antuñano, José María Romo de los Reyes, María del Pilar Sanjuán Cárdenas, Luis Serrano Cogollor, Ana E. del Villar Vázquez

**Affiliations:** 1Gynecologic Oncology, Private Practice (Laboratorio citología Dr. Cortés), 07004 Palma, Spain; 2MD Anderson Cancer Center, 28033 Madrid, Spain; 3Cervical Pathology Unit, Gynaecology Department, Hospital Sanitas La Zarzuela (Madrid), 28006 Madrid, Spain; 4Clínica Ginecológica Women’s CD, 08017 Barcelona, Spain; damiandexeus@gmail.com; 5Clínica Alborán CMM, 04001 Almería, Spain; gabrielfiol@clinicaalboran.es; 6Centro Ginecológico de León, 24004 León, Spain; 7HM Gabinete Velázquez, 28001 Madrid, Spaineleesece@yahoo.es (L.S.C.); 8Oncologic Gynecology, Lower Genital Tract Unit, Women’s Health Area, Hospital Universitari i Politécnic La Fe Valencia, 46026 Valencia, Spain; gurrea_mar@gva.es; 9Lower Genital Tract & Endoscopic Gyneacology Unit, Hospital Universitario Infanta Leonor, 28031 Madrid, Spain; emlamela@hotmail.com; 10Instituto Palacios de Salud de la Mujer, 28009 Madrid, Spain; 11Obstretrics and Gynaecology Unit, Hospital Universitario Virgen de Valme, 41014 Sevilla, Spain; jmromo901@hotmail.com; 12Clínica GINEMED, 41010 Sevilla, Spain; 13Clínica Millenium-Dent, 28005 Madrid, Spain; alexapons@gmail.com

**Keywords:** cervical lesions, *Coriolus versicolor*, human papillomavirus, real life, vaginal gel

## Abstract

**Simple Summary:**

Human papillomavirus (HPV) is responsible for virtually all cervical cancers in women. Although in the case of low-grade lesions the conventional approach is watchful waiting, some infections may lead to different-grade squamous intraepithelial lesions that can result in high-grade lesions and cancer. In this context, Papilocare^®^ is a vaginal gel that combines components with moisturizing, tissue regeneration, and microbiota-balancing properties with ingredients that have positive effects on HPV-dependent cervical lesions and HPV clearance. According to the results obtained in this observational study with real-world data, most participants repaired their cervical lesions and cleared HPV through Papilocare^®^ treatment, with a good treatment tolerability, adherence, and satisfaction rates, regardless of being infected with high-risk HPV, age over 40 years or both of them. Data reported in this study strengthen the role of the vaginal gel Papilocare^®^ as a complementary action to the simple watchful waiting approach in HPV-related low-grade cervical lesions management.

**Abstract:**

Human papillomavirus (HPV) is responsible for virtually all cervical cancers in women. HPV infection and persistency may lead to different-grade squamous intraepithelial lesions that can result in high-grade lesions and cancer. The objective was to prospectively evaluate the results of using a *Coriolus-versicolor*-based vaginal gel (Papilocare^®^) on HPV-dependent low-grade cervical lesion repair in a real-life scenario. HPV-positive women ≥ 25 years with ASCUS/LSIL cervical cytology results and concordant colposcopy images were included, receiving the vaginal gel one cannula/day for 21 days (first month) + one cannula/alternate days (five months). A 6-month second treatment cycle was prescribed when needed. Repair of the cervical low-grade lesions through cytology and colposcopy, HPV clearance, and level of satisfaction, and tolerability were evaluated. In total, 192 and 201 patients accounted for the total and safety analyses, respectively, and 77.1% repaired cervical lesions at 6 or 12 months (76.0% for high-risk HPV). Additionally, 71.6% achieved HPV clearance throughout the study’s duration (70.6% for high-risk HPV). Satisfaction level was rated 7.9 and 7.5 out of 10 at 6 and 12 months, respectively. Only three mild–moderate product-related adverse events were reported, and all of them were resolved by the end of the study. In our study, we observed higher regression rates of low-grade cervical lesions in women treated with Papilocare^®^ vaginal gel than spontaneous regression rates reported in the literature.

## 1. Introduction

Human papillomavirus (HPV) is the most common sexually transmitted infection, with most sexually active men and women acquiring it during their lifetime [1]. More than 200 types of HPV have been identified, 14 of them classified as high-risk (HR) variants, which are the main cause of virtually all precancerous cervical lesions and cervical cancers in women [2]. Most HPV infections and related low-grade cervical lesions regress spontaneously [3,4]. However, some infections become persistent. Some determinants for a persistent infection include HPV genotype, viral load, age at the time of detection, presence of vaginal dysbiosis, and low-immunity situations [5,6,7,8]. The persistence of the virus (especially HR HPV variants) is strongly linked to progression to more advanced precancerous lesions and, eventually, to cancer [9,10]. Regarding HPV genotype, the risk of progression to cervical intraepithelial neoplasia (CIN) 3 lesions within 5 years is considerably higher in HR variants (20.5%) than in low-risk ones (1.7%) [11]. When considering age, spontaneous regression rates of CIN lesions diminish from 44.7% in women under 25 to 24.9% in patients over 40 [12].

Due to the severe burden linked to this virus, therapies that deal with persistent HPV infection are essential. Despite the efficacy of vaccines in reducing HPV infections, these prophylactic and preventive methods are not effective in targeting already-established infections and HPV-induced neoplasms [13]. On the other hand, the conventional management of this condition, especially in cases of low-grade lesions, is watchful waiting along with other forms of guidance, such as maintaining good life habits and ceasing tobacco use if applicable. This approach may become long and challenging for both women and their physicians [14]. Many women experience negative emotional responses and even long-term psychological distress linked to an HPV-positive diagnosis and abnormal cytology, which could lead them to require additional psychological support [15,16]. Consequently, a safe and nonsurgical treatment that can repair low-grade lesions and enhance viral clearance might be highly useful. Such a procedure should prevent progression while decreasing the rate of conizations [9,17].

In this context, Papilocare^®^ (Procare Health, Spain) is a vaginal gel that combines components with moisturizing, tissue regeneration, and microbiota-balancing properties (hyaluronic acid, *Centella asiatica*, *Aloe vera* and α-glucan oligosaccharide) [18,19] with ingredients that have positive effects on HPV-dependent cervical lesions and HPV clearance (*Coriolus versicolor*, *Azadirachta indica* and carboxymethyl-β-glucan) [20,21,22]. It has proven to have a positive effect and safety in repairing low-grade cervical lesions and enhancing HPV clearance in studies of various designs [14,17]. However, thus far, no multicenter prospective studies have been performed regarding the outcomes of the use of this gel in a real-world setting. Therefore, the main objective of the present study was to prospectively evaluate the results of using Papilocare^®^ in repairing low-grade cervical lesions induced by HPV in routine clinical practice conditions.

## 2. Material and Methods

### 2.1. Study Design

This was an observational, national, multicentric, prospective, non-comparative clinical study performed between May 2018 and February 2021 (PAPILOBS trial, National Clinical Trial Number; #NCT04199260; www.clinicaltrial.gov (accessed on 12 January 2022); Appendix A).

Patients were recruited from public and private gynecology centers and services in Spain with experience in the follow-up of patients with cervical lesions due to HPV. Inclusion criteria included: women over 25 years old, vaccinated or not against HPV; an HPV-positive test of, at most, three months prior to the study inclusion; cervical atypical squamous cells of undetermined significance (ASCUS) or low-grade squamous intraepithelial lesion (LSIL) cytology result of, at most, three months prior to the study inclusion together with colposcopy image showing similar level of cervical dysplasia, which was considered concordant with cytology test. To measure the level of satisfaction, visual analogue scale (VAS) was used in patients prescribed Papilocare^®^ based on medical decision prior to their participation in the study. On the other hand, exclusion criteria included: any situation/alteration/pathology, gynecological or not, which contraindicated the use of Papilocare^®^, as it should not be used in case of hypersensitivity to any of its components [23]. Another exclusion criteria included women of childbearing age who did not use effective contraceptive methods; pregnant women, suspected pregnancy, or desire to become pregnant within the next six months; breastfeeding; participation in any other clinical trial at the time or in the four weeks prior to inclusion in the study; any planned surgery that precluded correct compliance with the guideline; and any known allergies to any of the components of Papilocare^®^. In addition, all patients were required to read the patient information sheet and to sign the informed consent form to be included in the study.

Participants were treated with Papilocare^®^ one cannula/day for 21 days during the first month, followed by one cannula/alternate days during the subsequent five months. All patients with normal cytology/colposcopy and HPV clearance ended the study after six months (second visit). Those who continued with altered cytology/colposcopy and/or HPV persistency were offered an additional 6-month treatment period with the same dosage, making their total treatment time 12 months (third visit).

This study was approved by the Ethics Committee of Puerta de Hierro Majadahonda University Hospital (Madrid, Spain). All procedures followed were in accordance with the Declaration of Helsinki and subsequent amendments.

### 2.2. Data Collection and Study Variables

The primary endpoint was the repair of the cervical low-grade lesions measured through the percentage of patients with normalized cervical cytology together with concordant colposcopy image. Secondary objectives included the HPV clearance, the level of satisfaction with the product, and the tolerability of Papilocare^®^. In addition, biopsy evolution was evaluated for those patients who had them available based on routine clinical practice. Improvement in the evolution was described as a change in the histological result in the following order—CIN-3, CIN-2, CIN-1—suggestive of inflammatory HPV and negative. The presence of HPV was measured through a PCR assay. HPV clearance was defined as a negative HPV test or disappearance of all strains present at baseline or disappearance of at least one baseline strain together with normal cytology findings and concordant colposcopy observations. A concordant colposcopy with ASCUS, LSIL, or AGUS cytological results were those findings classified as normal or abnormal in grade 1 (minor) according to the definition from the committee of the International Federation of Cervical Pathology and Colposcopy (IFCPC), defined at the Rio World Congress, 5 July 2011 [24]. Regarding the subtypes, those considered as HR are: 16, 18, 31, 33, 35, 39, 45, 51, 52, 56, 58, 59, and 68. To measure the level of satisfaction, a VAS that comprised values from 0 (not satisfied) to 10 (very satisfied) was used. The patient self-assessed the satisfaction level during each visit to the center. Adherence to the prescribed treatment was assessed via direct question to the patient in each study visit. Incidence, nature, severity, and relation with the treatment of adverse events were collected using a case report form (CRF) by the principal investigators (PI’s) and/or collaborators of each center during each visit by direct question. Patients received a patient information sheet with the contact details of the PI of the center or some of their collaborators for any incident potentially related to the study. Both primary and secondary objectives were evaluated after 6 months of treatment, after 12 months if relevant, and overall (6 or 12 months).

All data from the clinical study were collected in the CRF; baseline data were gathered during the recruitment visit either via direct question to the patient or during the medical exploration. Data obtained from 6- and 12-month visits were collected in the same way as the baseline data.

### 2.3. Sample Size Determination and Data Analysis

A sample size of 263 patients provided a ±4.95% precision to identify 80% of patients with their cervical lesions repaired with a 95% confidence interval and assuming 5% losses. Two patient samples were established. The total sample included all patients who met inclusion criteria and with both baseline data and any value corresponding to the primary endpoint. On the other hand, any patient who participated in the study and had at least one application of the investigational product was included in the safety sample.

Categorical variables were expressed as absolute and relative frequencies, while continuous variables were represented as mean and standard deviation (SD), 95% confidence interval, median, 25th and 75th percentiles, and minimum and maximum, including the total number of valid values.

In the case of qualitative variables, comparisons among patient subgroups and visits were performed through the chi-square and McNemar tests, respectively. Additional analyses of both the primary and HPV clearance endpoints were made with data obtained from 3 different subpopulations: patients with HR HPV, patients over 40 years old, and patients over 40 with HR HPV. Statistical significance was established at *p* ≤ 0.05. All statistical procedures were performed using SAS v9.4.

## 3. Results

Out of 263 enrolled patients, 192 and 201 accounted for the total and safety samples, respectively (Figure 1).

### 3.1. Baseline Characteristics

The mean age in the total sample was 38.7 ± 9.1 years, with 95% of them being Caucasian. Thirty-three patients (17.6%) were vaccinated against HPV, with a 4.3 ± 4.6-year mean time since its administration. Regarding sexual behavior patterns, the participants had a mean number of partners and intercourses within the last month of 1.0 ± 0.5 and 6.3 ± 5.4, respectively. Regarding clinical data, 58.9% and 41.1% of patients had LSIL and ASCUS results in the pap smear, respectively, and 93.2% had HR HPV. Other baseline characteristics are observed in Table 1.

### 3.2. Primary Endpoint: Repair of HPV-Induced Cervical Lesions

Overall, by the end of the study, the vaginal gel managed to repair cervical lesions in 77.1% of the participants. After 6 months of treatment, 67.0% of the participants had their cervical lesions repaired (95% CI: 60.4–73.7%). Among women that did not repair their lesions in the first 6 months, 58.8% did so by 12 months (95% CI: 42.3–75.4%). The repairing of lesions was achieved in 76% (95% CI: 69.7–82.2%) of patients infected with the HR HPV. At the second and third visits, 66.9% (95% CI: 59.9–73.8%) and 54.8% (95% CI: 37.3–72.4%) of patients had a normalized cytology with a concordant colposcopy image, respectively. The sub-analysis of women over 40 showed that 82.4% (95% CI: 73.8–91.1%) had their lesions repaired overall (74.0% (95% CI: 63.9–84.0%) and 58.3% (95% CI: 30.4–86.2%) in the second and third visits, respectively). Regarding patients over 40 infected with HR HPV, 81.2% (95% CI: 71.9–90.4%) had their lesions repaired overall. At the second and third visits, 73.5% (95% CI: 63.0–84.0%) and 54.5% (95% CI: 25.1–84.0%) reported a normalized cytology with a concordant colposcopy image, respectively (Table 2).

### 3.3. HPV Clearance

A total of 71.6% of patients achieved HPV clearance by the end of the study. After 6 months of treatment, HPV test results showed 58.7% clearance (95% CI: 51.7–65.8%). Among women that did not achieve HPV clearance in the first 6 months and continued treatment for a further 6 months, 52.1% (95% CI: 38.0–66.2%) achieved it at 12 months. For these patients, data were not available from 30 of them, which includes the patients who did not continue in the study after visit 2 by their own or their physician’s decision, even though they had not cleared HPV (n = 18), and those who were lost/dropped out between visits 2 and 3 (n = 12), from which five were lost during the follow-up visits—three left due to their physician’s decision, one withdrew her consent, and the situations of three were unknown. In participants infected with the HR HPV, HPV clearance was achieved overall by 70.6% (95% CI: 63.9–77.3%) of patients, while 57.4% (95% CI: 50.1–64.7%) and 52.2% (95% CI: 37.7–80.1%) achieved it at 6 and 12 months, respectively. In the case of women over 40, HPV clearance was 75.3% (95% CI: 65.5–85.2%) overall (61.1% (95% CI: 49.9–72.4%) and 57.9% (95% CI: 35.7–80.1%) at the second and third visits, respectively). Regarding patients over 40 infected with HR HPV, 73.5% (95% CI: 63.0–84.0%) of women managed to clear the HPV. At the second and third visits, clearance had occurred in 59.7% (95% CI: 48.0–71.5%) and 55.6% (95% CI: 32.6–78.5%) of patients, respectively (Figure 2).

### 3.4. Biopsy Results

A total of 91 patients had a baseline biopsy taken within 3 months before starting the treatment, of which CIN-1 was the most frequent histological result, representing 61.5% of the analysed cases. Among patients with a biopsy performed at visit 2 (n = 26) and visit 3 (n = 13), 76.9% and 69.3%, respectively, did not show changes in the biopsy results vs. baseline. However, at both follow-up visits, more biopsy improvements than worsening vs. baseline were observed: 15.4% (4 CIN-1 to 3 negative and 1 inflammatory) vs. 7.2% (2 inflammatory to 1 suggestive of HPV and 1 CIN-1) at visit 2 and 23.5% (3 CIN-1 to negative) vs. 7.7% (1 inflammatory to CIN-1) at visit 3 (Appendix A).

### 3.5. Satisfaction, Adherence, and Tolerability with Papilocare^®^

On a 0 to 10 scale, the satisfaction level with the investigation product remained between 7.5 and 8.0 in both visits. While on the second visit, the mean score was 7.9 ± 1.8, patients rated the product with a 7.5 ± 1.9 on the third visit. In both visits, most of the participants gave a score of 5 out of 10 or above to the product (93.7% and 95.1% in visits 2 and 3, respectively), with 42.4% and 31.1% of women, respectively, giving scores of 9 or higher in each visit (Table 3). Adherence to the prescribed treatment was assessed via direct question and was very high after 6 months of treatment (visit 2)—94.2% of patients were compliant. Remarkably, after 12 months of treatment (visit 3), the compliance with the treatment remained very high, with an adherence rate of 98.4%. However, this value might be overestimated because the 12 patients who lost/dropped out between visits 2 and 3 due to any of the reasons mentioned in Section 3.3. These patients might be considered as having a poor adherence to treatment.

A total of eight patients reported eight adverse events (AEs), with only three of them considered to be related to the product under investigation. These AEs included two mild and one moderate feeling of vulvovaginal itching/stinging. All of them were resolved by the end of the study.

## 4. Discussion

Results of this real-life study aligned with previously published data, in which Papilocare^®^ emerges as an effective and safe option for the treatment of HPV-induced low-grade cervical lesions [14,17]. HPV and low-grade lesions usually clear and regress spontaneously [3,4]. Due to this and the lack of effective conservative treatments, the most common management approach is watchful waiting, which can cause significant stress and anxiety in many women, which could—in turn—provoke a diminished immune response. On the other hand, systematic reviews have established that the excisional and ablative treatment approach for CIN is conflicting for specialists, since excisional techniques increase the risk of subsequent preterm birth, and ablative treatments are linked to higher failure rates [24]. Although these therapies might lead to undesired reproductive outcomes, around 10% of the cases progress to higher grade lesions if treatment is avoided [25]. In addition, recommendations from Spanish guidelines and health authorities, like the World Health Organization, include the disease monitoring [26,27,28].

In this context, Papilocare^®^, a *Coriolus*-*versicolor*-based vaginal gel which has shown to improve the epithelialization of cervical mucosa and the composition of vaginal microbiota emerges as a safe treatment option for helping to repair low-grade lesions and clear HPV and, additionally, may help lower stress levels in women by making them active participants in their own treatment [29,30]. The efficacy of the vaginal gel has been tested in a controlled clinical trial [14] in which, after 6 months of treatment, significantly higher percentage of treated patients achieved a normal cytology with a concordant colposcopy in comparison to the control group (84.9% vs. 64.5%; *p* = 0.031). Regarding HR HPV clearance, the superiority of the vaginal gel in the randomized trial was also reported at 6 months, although only a statistical trend was observed (62.5% vs. 40.0%). The results were slightly lower in this real-life study (67.0% lesion repair and 58.7% for HR HPV clearance at six months) although these two studies are difficult to compare due to the different methodological designs. While PALOMA is a clinical trial in which experiments were conducted under controlled conditions, PAPILOBS is a real-world study in which variables are not controlled. However, these studies are complementary, as efficacy and safety found under ideal conditions (such as in the PALOMA study) may substantially differ from those found under real-world settings [31]. Despite the lack of a control group, the normalization rates were higher than those spontaneously achieved described in the literature. This was a study with a similar design and mean age population to ours and included 570 women with LSIL cytology with a mean age of 36.0 years (similar cohort to our study, in which the average was 38.7 years); the cumulative probability of regression in the 2-year follow-up was 62.3% [11]. Furthermore, in observational cohorts, it is estimated that 35–40% of women normalize their CIN-1 lesions within a year, with 40–65% of them achieving that goal within two years [32,33,34]. In contrast, the normalization rates reported in the PAPILOBS study were 67% at 6 months and 77.1% at 12 months. In addition, age deeply impacts both the regression and clearance rates. In a longitudinal study with a large cohort, spontaneous regression rates of CIN lesions diminish from 44.7% in women under 25 to 24.9% in patients over 40 [12]. In our study, 81.2% and 59.7% of women over 40, described as a population sensitive to the virus persistence due to age-related immunosenescence, had their lesions normalized and HPV cleared, after 6 months. In addition, the 6-month positive effect of Papilocare^®^ has been addressed by several independent retrospective observational studies [17,35,36]. Criscuolo et al. [17] showed that 6 months after starting a 3-month treatment with Papilocare^®^, the HPV DNA test became negative in 67.0% versus 37.2% of the controls (*p* < 0.0001). Furthermore, 76.1% versus 40.8% registered a colposcopy improvement (*p* = 0.0005), and 60.4% versus 40.8% showed a remission (*p* = 0.05) for treated versus controls, respectively, in 183 HR-HPV-positive patients. Gajino Suárez [35] reported 58% HPV clearance rates in women infected by HR HPV, increasing to 64% in women over 65. Finally, the tolerability profile was similar in all studies, characterized by the lack of severe AEs as well as the high satisfaction with the product reported by participants. Compared to the PALOMA study, adherence was similar after six months of treatment (94.2% vs. 94.3%), as was tolerability profile (three vs. seven AEs were considered related to the product under investigation). In the case of adherence, the results were high considering the long periods of product application and frequent uses of the product under investigation (daily or on alternate days), which shows the awareness of patients towards this pathology.

The prospective design with a considerable number of evaluated patients is the main strength of this observational study. On the other hand, limitations included the absence of a control arm and the high number of losses due to the COVID pandemic. In addition, there is a lack of information about both cofactors relevant for the progression of cervical HPV infection to cancer (such as smoking) and covariable-adjusted analyses, which could have avoided potential interferences in the results [37]. The lack of an analysis of regression and clearance for specific HPV genotypes can also be considered a limitation. Another limitation is the inclusion of the “disappearance of at least one baseline strain together with normal cytology findings and concordant colposcopy observations” as an HPV clearance endpoint. This decision was due to the idea that one viral subtype frequently causes one HPV-related lesion regardless of the presence of multiple HPV types [38]. However, with both the regression and clearance of one of the oncogenic HPV subtypes, it is likely that the elimination of the causative virus has been performed. Nevertheless, the risk of future lesions and dysplasia remains if the residual HPV subtype is of an HR subtype. Lastly, a definitive conclusion about improvements regarding the lesion type cannot be made due to the low levels of biopsies analyzed.

## 5. Conclusions

In our study we observed higher regression rates of low-grade cervical lesions in women treated with the vaginal gel Papilocare^®^ than the spontaneous regression rates reported in the literature. In addition, it has shown an adequate tolerability profile, with high satisfaction and adherence rates. These results are in line with those observed in the PALOMA clinical trial and in other observational studies, which strengthens the role of this vaginal gel as a beneficial alternative to the simple watchful waiting approach in HPV-related low-grade cervical lesions.

## Figures and Tables

**Figure 1 cancers-15-03863-f001:**
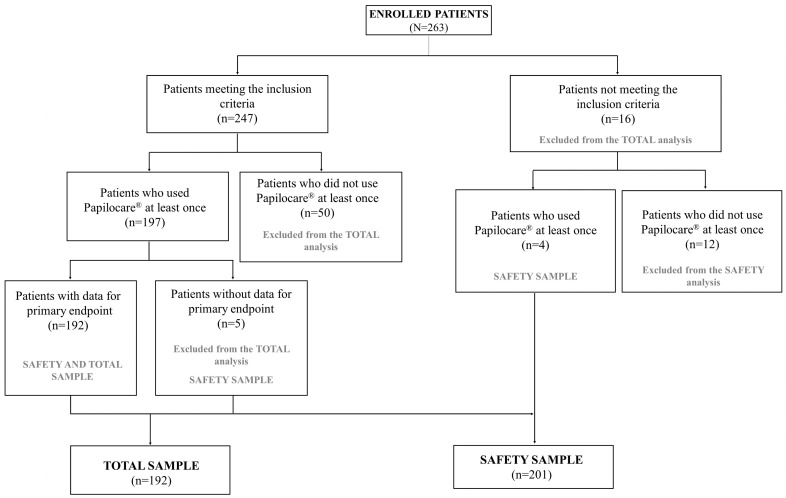
Patient flow chart.

**Figure 2 cancers-15-03863-f002:**
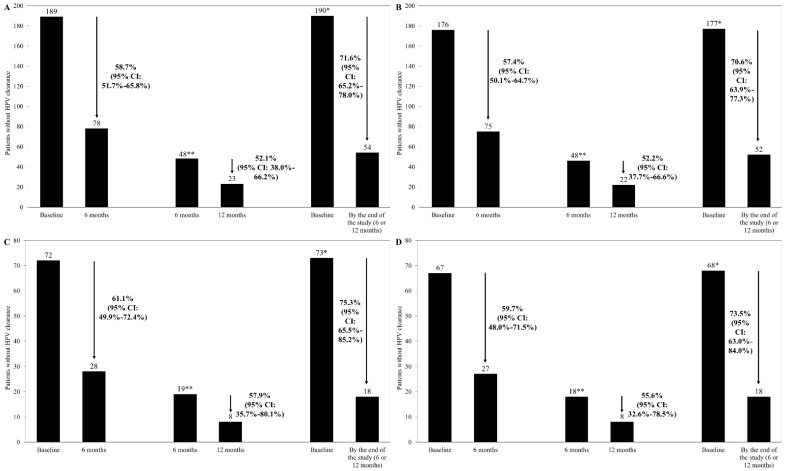
HPV clearance rates in the overall population (**A**), patients infected with high-risk variants of the virus (**B**), patients over 40 (**C**), and patients over 40 and with high-risk HPV (**D**) after 6 and 12 months of treatment and by the end of the study (after 6 or 12 months). * Number of patients based on full treatment period (one patient was not HPV tested at the 6-month visit but was tested at the 12-month visit): ** Patients included in the additional 6-month treatment period and with data from 12-month visit.

**Table 1 cancers-15-03863-t001:** Baseline characteristics of the total sample (*n* = 192).

Age, Mean Years (SD)	38.7 (9.1)
Ethnicity, *n* (%)	
Caucasian	182 (95.3)
Other (Maghreb, Afro-American, Latin-American, Asian)	9 (4.7)
BMI, mean kg/m^2^ (SD)	22.9 (3,7)
Relevant disease/surgery, *n* (%) ^1^	189 (100.0)
Yes	15 (7.5)
Specifications of relevant disease/surgery, *n* (%) ^2^	16 (100.0)
Infections/Infestations	1 (6.3)
Benign/malignant/non-specified neoplasia	4 (25.0)
Medical/surgical procedures	3 (18.8)
Cardiac disorders	1 (6.3)
Skin and subcutaneous tissue disorders	1 (6.3)
Endocrine disorders	1 (6.3)
Gastrointestinal disorders	2 (12.5)
Psychiatric disorders	1 (6.3)
Vascular disorders	2 (12.5)
Relevant gynecological disease/surgery, *n* (%) ^3^	187 (100.0)
Yes	15 (8.0)
Specifications of relevant gynecological disease/surgery, *n* (%) ^4^	17 (100.0)
Benign/malignant/non-specified neoplasia	2 (11.8)
Medical/surgical procedures	11 (64.7)
Reproductive system/breast disorders	4 (23.5)
Sexual behavior pattern	
Number of partners within the last month, mean (SD)	1.0 (0.5)
Number of intercourses within the last month, mean (SD)	6.3 (5.4)
Use of condoms within the last month, *n* (%)	
Always	63 (32.8)
Sometimes	61 (31.8)
Never	38 (19.8)
Not applicable ^5^	30 (15.6)
HPV vaccine, *n* (%)	192 (100.0)
Yes	33 (17.6)
No	139 (74.3)
Unknown	20 (8.0)
Years since HPV vaccine administration, mean years (SD)	4.3 (4.6)
Cytology, *n* (%)	
Atypical squamous cells of undetermined significance	79 (41.1)
Low-grade squamous intraepithelial lesion	113 (58.9)
HPV test, *n* (%)	
High risk	179 (93.2)
Low risk	10 (5.2)
Unknown	3 (1.6)
Biopsy, *n* (%)	
Yes	91 (47.4)
No	101 (52.6)
Results from the biopsy, *n* (%) ^6^	
Negative	12 (13.2)
Suggestive of inflammatory HPV	15 (16.5)
CIN-1	56 (61.5)
CIN-2	5 (5.5)
CIN-3	3 (3.3)
Concomitant medication, *n* (%)	
Yes	10 (5.2)
No	182 (94.8)

BMI, body mass index; HPV, human papillomavirus; SD, standard deviation. ^1^ *n* = 3 patients with no general history information. ^2^ *n* = 15 patients specified *n* = 16 relevant diseases/surgeries. Percentages calculated from total specified disease/surgeries (*n* = 16). ^3^ *n* = 5 patients with no gynecological history information. ^4^ *n* = 15 patients specified *n* = 17 relevant gynecological diseases/surgeries. Percentages calculated from total specified relevant gynecological diseases/surgeries (*n* = 17). ^5^ In case the patient had no partner and/or no sexual intercourse within the last month. ^6^ Percentages based on the total number of patients with biopsy (*n* = 91).

**Table 2 cancers-15-03863-t002:** Cytology and colposcopy results observed after 6 months (visit 2), 12 months (visit 3), and overall (6 or 12 months) with Papilocare^®^ in the total sample, patients with HR HPV, patients over 40 and patients over 40 infected by HR HPV.

	Total Sample			HR HPV			Women over 40			Women over 40 + HR HPV		
	Visit 2 (*n* = 191) ^1^	Visit 3 (*n* = 34) ^2^	Visit 2 or 3 (*n* = 192) ^3^	Visit 2 (*n* = 178) ^1^	Visit 3 (*n* = 31) ^2^	Visit 2 or 3 (*n* = 179) ^3^	Visit 2 (*n* = 73) ^1^	Visit 3 (*n* = 12) ^2^	Visit 2 or 3 (*n* = 74) ^3^	Visit 2 (*n* = 68) ^1^	Visit 3 (*n* = 11) ^2^	Visit 2 or 3 (*n* = 69) ^3^
Cytology result + colposcopy, n (%)												
Normalization	130 (68.1)	21 (61.8)	151 (78.6)	121 (68.0)	18 (58.1)	139 (77.7)	54 (74.0)	7 (58.3)	61 (82.4)	50 (73.5)	6 (54.5)	56 (81.2)
With concordant colposcopy ^4^	128 (98.5)	20 (95.2)	148 (98.0)	119 (98.3)	17 (94.4)	136 (97.8)	54 (100.0)	7 (100.0)	61 (100.0)	50 (100.0)	6 (100.0)	56 (100.0)
With dissenting colposcopy ^4^	2 (1.5)	1 (4.8)	3 (2.0)	2 (1.7)	1 (5.6)	3 (2.2)	0 (0.0)	0 (0.0)	0 (0.0)	0 (0.0)	0 (0.0)	0 (0.0)
Abnormal	61 (31.9)	13 (38.2)	41 (21.4)	57 (32.0)	13 (41.9)	40 (22.3)	19 (26.0)	5 (41.7)	13 (17.6)	18 (36.5)	5 (45.5)	13 (18.8)
Repair of lesions (normalized cytology with concordant colposcopy image), *n* (%)												
Yes	128 (67.0)	20 (58.8)	148 (77.1)	119 (66.9)	17 (54.8)	136 (76.0)	54 (74.0)	7 (58.3)	61 (82.4)	50 (73.5)	6 (54.5)	56 (81.2)
No	63 (33.0)	14 (41.2)	44 (22.9)	59 (33.1)	14 (45.2)	43 (24.0)	19 (26.0)	5 (41.7)	13 (17.6)	18 (26.5)	5 (45.5)	13 (18.8)

^1^ Number of patients with data from visit 2. ^2^ Number of patients without lesions repaired in visit 2 and with available data from visit 3. ^3^ Number of all evaluated patients (*n* = 1 patient did not attend visit 2, but data from visit 3 were retrieved). It comprised data from patients with normalized cytology with concordant colposcopy image after 6 months of treatment, and data from participants without a normalized cytology with concordant colposcopy image in visit 2 that attended visit 3. ^4^ Percentage values calculated from patients with a normal cytology result in each visit. HR: high-risk; visit 2: 6 months; visit 3: 12 months.

**Table 3 cancers-15-03863-t003:** Level of satisfaction after 6 months (visit 2) and 12 months (visit 3) of treatment with Papilocare^®^.

Satisfaction Scale *	Visit 2 (*n* = 191)	Visit 3 (*n* = 61)
Mean score (SD)	7.9 (1.8)	7.5 (1.9)
Patients n (%)		
≥5 points	179 (93.7)	58 (95.1)
≥7 points	153 (80.1)	47 (77.0)
≥9 points	81 (42.4)	19 (31.1)

* Visual analogue scale from 0 (not satisfied) to 10 (very satisfied).

## Data Availability

The data that support the findings of this study are available from the corresponding author upon reasonable request.

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
