# Peer review of "Effect of a Multi-Ingredient Coriolus-versicolor-Based Vaginal Gel in Women with HPV–Dependent Cervical Lesions: The Papilobs Real-Life Prospective Study"

_cancers, 2023, doi:10.3390/cancers15153863_

Round 1
Reviewer 1 Report
The main objective of the present study was to prospectively evaluate the effectiveness and tolerability of the vaginal gel Papilocare® in repairing cervical low-grade lesions induced by HPV in routine clinical practice conditions. The authors conclude that this real-life study strengthen previously published data, in which Papilocare® emerges as an effective and safe option for the treatment of HPV-induced low-grade cervical lesions.
Although the study is of interest there are several serious concerns:
1. The title of the manuscript is misleading. In order to determine the efficacy of any treatment a control arm is required!!! The control arm is lacking in this study.
2. The authors claim that data obtained in the current study are similar to the data obtained in the previously conducted PALOMA study in which a control arm was used. However, in the PALOMA study two different Papilocare treatment schemes were compared with untreated controls. Treatment scheme A of the PALOMA study is identical with the treatment scheme of the current study for the first 6 months. Notably, in the PALOMA study only scheme B showed significant differences in HPV clearance compared with the control group (75.9% vs 41.9%, p = .008) after 6 months, whereas no significant differences were observed with scheme A (39.1% vs 41.9%, p = .836). In view of this poor performance of treatment scheme A, the rational for choosing schema A for the current study is not clear.
Moreover, with regard to clearing/repair of lesions after treatment, in the PALOMA study the percentage of patients presenting with normal Pap smear findings and concordant colposcopy observations after 6 months was significantly higher in the treatment group than in the control group (84.9% vs 64.5%, p = .031). However, for this analysis the data of patients of both treatment schemes were combined. Thus, the effectivity of clearing/repair of lesions with treatment scheme A was not disclosed. In the light of the findings of the PALOMA study, the current study does not provide convincing evidence that treatment with Papilocare is really effective.
3. Other than for the PALOMA study, treatment in the current study was continued for up to 12 months. Evidently, prolonged treatment has a favourable effect on HPV clearance and lesion repair. However, without a control arm it cannot be claimed that this effect is related to the treatment. It is well known that most low grade lesions resolve spontaneously over time. The increase in HPV clearance and lesion repair over time is thus not unexpected.
Author Response
Dear Editor, Cancers
Enclosed please find the revised version of the manuscript, REAL-LIFE EFFECTIVENESS OF A MULTI-INGREDIENT CORIOLUS VERSICOLOR-BASED VAGINAL GEL IN WOMEN WITH HPV–DEPENDENT CERVICAL LESIONS: THE PAPILOBS PROSPECTIVE STUDY, for your consideration for publication in Cancers.
In the following pages, we have addressed the two reviewers’ comments point by point. We hope this revised version of the manuscript is now suitable for publication.
Thank you in advance for considering this manuscript. We look forward to hearing from you.
Cordially,
Javier Cortés Bordoy, MD, PhD.
Reviewer 1:
Comments and Suggestions for Authors
The main objective of the present study was to prospectively evaluate the effectiveness and tolerability of the vaginal gel Papilocare® in repairing cervical low-grade lesions induced by HPV in routine clinical practice conditions. The authors conclude that this real-life study strengthen previously published data, in which Papilocare® emerges as an effective and safe option for the treatment of HPV-induced low-grade cervical lesions. Although the study is of interest there are several serious concerns:
The title of the manuscript is misleading. In order to determine the efficacy of any treatment a control arm is required!!! The control arm is lacking in this study.
ANSWER: The title is now modified as suggested. Other minor change has been done in tittle: “Effect of a multi-ingredient Coriolus versicolor-based vaginal gel in women with HPV–dependent cervical lesions: the Papilobs real-life prospective study.” (lines 1-5) The term “efficacy” has been substituted by “effect”, as we agree with the fact that efficacy is measured in randomized controlled trials (RCT). The efficacy of the Coriolus versicolor-based vaginal gel was previously evaluated in the PALOMA RCT[1], in which a control arm was included. In this single-arm observational study we describe the effect in a given cohort representing the general population in real life conditions, which could potentially deviate significantly from the previously reported data in the PALOMA RCT. Some of these relevant differences may come from the population, which is usually more heterogeneous in the observational studies, the number of patients, the status of the pathology, how the product is applied, the adherence or even how the product is stored.
The authors claim that data obtained in the current study are similar to the data obtained in the previously conducted PALOMA study in which a control arm was used. However, in the PALOMA study two different Papilocare treatment schemes were compared with untreated controls. Treatment scheme A of the PALOMA study is identical with the treatment scheme of the current study for the first 6 months. Notably, in the PALOMA study only scheme B showed significant differences in HPV clearance compared with the control group (75.9% vs 41.9%, p = .008) after 6 months, whereas no significant differences were observed with scheme A (39.1% vs 41.9%, p = .836). In view of this poor performance of treatment scheme A, the rational for choosing schema A for the current study is not clear.
Moreover, with regard to clearing/repair of lesions after treatment, in the PALOMA study the percentage of patients presenting with normal Pap smear findings and concordant colposcopy observations after 6 months was significantly higher in the treatment group than in the control group (84.9% vs 64.5%, p = 0.031). However, for this analysis the data of patients of both treatment schemes were combined. Thus, the effectivity of clearing/repair of lesions with treatment scheme A was not disclosed. In the light of the findings of the PALOMA study, the current study does not provide convincing evidence that treatment with Papilocare is really effective.
ANSWER: We agree with the fact that the PALOMA study had two different administration schemes included in the study. However, in order to accomplish the primary endpoint schemes A and B were pooled into a single-treatment group as stated in Paloma study publication: “Because evaluating the efficacy of Papilocare versus the control approach was the primary objective of the trial, data from treatment schemes A and B were pooled into a single-treatment group for the main analysis.” Considering that repair of the lesions was the primary endpoint in this study, the sample size was calculated according to this objective (in which both scheme, A and B, got statistically significant results in high-risk HPV population, data not shown in PALOMA publication).
Thus, despite the scheme B from the Paloma study was the one that showed significant results regarding the efficiency of clearance, HPV clearance was one of the secondary endpoints, and therefore it might be possible that the lack of statistical significance is due to a lack of statistical power, not lack of effect. Hence, the chosen scheme could not be based on the outcome obtained from a sub-analysis of secondary endpoint which might be affected by the lack of statistical power.
Additionally, both the RCT Paloma study and this observational study were performed at thoroughly at same time, and partially overlapping, so the results of the PALOMA study were unknown by the time in which PAPILOBS protocol was designed. Moreover, it was considered that scheme A would have an easier use and therefore, it would have better adherence rates in clinical practice using a minimum effective dose of Papilocare® treatment. All the previous reasons support why we included the scheme A for this study.
- Other than for the PALOMA study, treatment in the current study was continued for up to 12 months. Evidently, prolonged treatment has a favourable effect on HPV clearance and lesion repair. However, without a control arm it cannot be claimed that this effect is related to the treatment. It is well known that most low-grade lesions resolve spontaneously over time. The increase in HPV clearance and lesion repair over time is thus not unexpected.
ANSWER: We agree that a significant percentage of low-grade lesions have spontaneous regression. Arguably, the longer the treatment time, the higher the chances to normalise/clear the lesions. Nonetheless, the regression rates are widely described in the literature. The study of 570 women with LSIL cytology with a mean age of 36.0 years (similar cohort to Papilobs that was 38.7 year-old on average) showed that the cumulative probability of regression in the 2-year follow-up was 62.3%[2]. Furthermore, it is estimated that 35-40% normalise CIN-1 lesions within a year and 40-60% in two years[3],[4],[5]. In contrast, the normalisation rates found in PAPILOBS are 67% at 6 months and 77.1% at 12 months.
Age deeply impacts both the regression and clearance rates. When considering age, spontaneous regression rates of CIN lesions diminish from 44.7% in women under 25 to 24.9% in patients over 40[6]. The control group of Paloma RCT achieved a 33% of normalization at 6 months in women above 40 years old[7]. Hence, we can infer that the regression rate observed in our cohort is higher than the one we would observe if these patients were just following the “wait and see” approach.
Regarding to clearance, patients from the PAPILOBS study achieved a clearance rate of 58.7% after 6 months and 71.6% after 12 months. It is well known that clearance rates are reduced with age, as it has been extensively reported in the literature[8],[9]. Thus, in the present study, clearance rates are clearly above what it would be expected for a cohort that is 38.7-year-old on average. Also, a 93.2% from the PAPILOBS cohort has high-risk (HR) HPV, while in the general population HR-HPV infections represents a much lower percentage[10],[11],[12],[13]. Hence, it can be speculated that our cohort would have lower HPV clearance rates than the cohorts normally described in the literature, as it is enriched in HR-HPV. However, we observed clearance rates that are clearly above what it would be expected for a cohort of these characteristics.
Age and HPV-genotype are the main factors that impact the regression/ persistence rates. In this study, we also provide data from women who are over 40-years-old and additionally, of those, the ones who carry HR-HPV.
To summarize, the regression (67% at 6 month and 77.1% at 12 months) and clearance rates (58.7% at 6 months and 71.6% at 12 months) are above the numbers reported in the literature. Our explanation to this observation is that the treatment with Papilocare might be behind these numbers. Nonetheless, we agree that this is a descriptive study, and we should we referring to this data as “the effect over the regression rates observed in this cohort of patients”, as a consequence we rephrase the manuscript to make it more accurate to what it can be discussed with the data provided in it.
[1] Serrano L, et al. Efficacy of a Coriolus versicolor-based vaginal gel in women with human papillomavirus-dependent cervical lesions: The PALOMA Study. J Low Genit Tract Dis 2021;25(2):130-136. doi: 10.1097/LGT.0000000000000596.
[2]Matsumoto K, et al. Predicting the progression of cervical precursor lesions by human papillomavirus genotyping: a prospective cohort study. Int J Cancer. 2011;128(12):2898-910. doi: 10.1002/ijc.25630.
[3]Dalstein V, et al. Persistence and load of high-risk HPV are predictors for development of high-grade cervical lesions: a longitudinal French cohort study. Int J Cancer. 2003;106(3):396-403. doi: 10.1002/ijc.11222.
[4]Bansal N, et al. Natural history of established low grade cervical intraepithelial (CIN 1) lesions. Anticancer Res. 2008;28(3B):1763-6.
[5]Melnikow J, et al. Natural history of cervical squamous intraepithelial lesions: a meta-analysis. Obstet Gynecol. 1998;92(4 Pt 2):727-35. doi: 10.1016/s0029-7844(98)00245-2.
[6]Bekos C, et al. Influence of age on histologic outcome of cervical intraepithelial neoplasia during observational management: results from large cohort, systematic review, meta-analysis. Sci Rep.;8(1):6383. doi: 10.1038/s41598-018-24882-2.
[7]Gil-Antuñano SP, et al. Efficacy of a Coriolusversicolor-Based Vaginal Gel in Human Papillomavirus-Positive Women Older Than 40 Years: A Sub-Analysis of PALOMA Study. J Pers Med. 2022;12(10):1559. doi: 10.3390/jpm12101559.
[8]Li W, et al. Association of age and viral factors with high-risk HPV persistence: A retrospective follow-up study. Gynecol Oncol. 2019;154(2):345-353. doi: 10.1016/j.ygyno.2019.05.026.
[9]Plummer M, et al. A 2-year prospective study of human papillomavirus persistence among women with a cytological diagnosis of atypical squamous cells of undetermined significance or low-grade squamous intraepithelial lesion. J Infect Dis. 2007;195(11):1582-9. doi: 10.1086/516784.
[10]Zheng LL, et al. High-risk HPV prevalence and genotype distribution among women in Liaocheng, Shandong Province, China from 2016 to 2022. Front Public Health. 2023;11:1145396. doi: 10.3389/fpubh.2023.1145396.
[11]He L, et al. Distribution of high-risk HPV types among women in Sichuan province, China: a cross-sectional study. BMC Infect Dis. 2019;19(1):390. doi: 10.1186/s12879-019-4038-8.
[12]Torres-Poveda K, et al. High risk HPV infection prevalence and associated cofactors: a population-based study in female ISSSTE beneficiaries attending the HPV screening and early detection of cervical cancer program. BMC Cancer. 2019;19(1):1205. doi: 10.1186/s12885-019-6388-4.
[13]Seyoum A, et al. Genotype heterogeneity of high-risk human papillomavirus infection in Ethiopia. Front Microbiol. 2023;14:1116685. doi: 10.3389/fmicb.2023.1116685.

Reviewer 2 Report
Summary
This article evaluates the effectiveness of a coriolus versicolor-based vaginal gel on ASC-US and LSIL cervical lesion repair and HPV clearance in women aged 25 years and older, as well as its safety and patient-satisfaction levels for its use. Although the paper is well written and results are clearly presented, the study design does not allow to answer the main objective of the product’s effectiveness on the repair of HPV-dependent cervical lesions as there is no control group included in the analysis.
General concept comments
1. Overall, the paper is well structured, the introduction provides a strong background and justification of the study, and the results are clearly presented. My main comments below are related to the methodology and interpretation of results.
2. The main issue in this paper is the choice of study design, specifically the absence of a control arm, to answer the objectives regarding effectiveness of the therapeutic product tested (repair of low-grade cervical lesions and HPV clearance). In its current form, the study design is only adequate to inform about acceptability/safety outcomes (taking into consideration however the relatively small sample size) and descriptive results of cytological/HPV infection outcomes.
3. Considering the previous point, the current conclusion is misleading as there is no way of knowing whether regression of ASC-US/LSIL and HPV clearance would have been lower without any treatment provided (which is currently the standard approach to low-grade lesions). In fact, previously published papers show similar results as presented in this paper for the natural regression of low-grade lesions (Schlecht et al: https://doi.org/10.1093/jnci/djg037) as well as for HPV clearance in women aged <40 years (Rodriguez et al: https://doi.org/10.1093/jnci/djq001 , Winer et al: https://doi.org/10.1158/1055-9965.EPI-10-1108).
4. The paper provides no information on the participants’ compliance to treatment, which would have been of interest considering the long period (6 to 12 months) and frequent applications (every day or on alternate days) of an intravaginal product. The study design would further be more appropriate to observe compliance than to assess effectiveness.
5. The generalizability of results is difficult to assess as insufficient information is provided regarding study sites and the procedure for participant selection/recruitment.
Specific comments
Abstract
1. Line 18: I would suggest changing the sentence to “Infection may lead to different grade…”
2. Line 26: “by the end of the study” à please clarify what this means.
3. The conclusion of the abstract is not supported by the study design (no control group - no evidence that HPV clearance/cervical lesion repair would have been lower with no treatment) (see further comments in conclusion section)
Introduction
1. The scientific background and rationale for the study are clearly explained.
2. References #9 & #10 (Hoffman et al. & Major et al.) do not support what is stated in the introduction at the place where they are cited (“The persistence of the virus is strongly linked to progression to more advanced precancerous lesions and, eventually to cancer [9,10].”)
3. Lines 59-60: “Such procedure should […] avoid future over-treatment” à please explain here what is meant by “avoid future over-treatment” (this should be better argumented as some may consider treating low-grade cervical lesions as “overtreatment”).
4. Line 63: “and other ingredients” à please detail or give a reference to the full description of the product used (ideally in the methods section).
5. Study main objectives are clearly stated. However, I would also recommend stating the secondary objective of evaluation HPV clearance.
Methods
1. The study type is well described. However, many other elements are missing in the methods section:
- What were the study sites?
- What was the recruitment procedure?
- Data collection: how was cytology obtained? Who conducted the analyses? Same for biopsy analyses.
- How was satisfaction assessed (self-completed questionnaire vs filled by a HCP, on-line vs in person? Timing of the assessment)?
- Evaluation of tolerability: how/when were side effects reported?
- Baseline characteristics: how were they obtained?
- Was follow-up part of the routine clinical care or only for the study? Describe exactly which tests were done at which visit (HPV, cytology, colposcopy done systematically at all 3 visits?)
- Which HPV types were considered high-risk (even if already mentioned in the introduction, please confirm that the same 14 types were considered high-risk in the analysis).
2. Line 75: #NCT04199260 à is this the clinicaltrial.gov number? Please specify.
3. Line 79: “with concordant colposcopy image” à please explain what is meant by this (how was it determined whether colposcopy was concordant with cytology, which criteria were used).
4. Line 81: please specify what the contraindications are to the use of the Papilocare gel.
5. Lines 92-97: this information should go in another section of the methods (e.g. study procedures), not data collection.
6. Lines 94-95: Please clarify. Did all partcipants with either normal cytology/colposcopy OR HPV clearance leave the study at 6 months? For example if cyto/colpo was pathological but the HPV was cleared, were they not followed up until 12 months? This also contradicts the following sentence on lines 95-96.
7. Line 105: which HPV PCR assay was used?
8. Line 116: you use here the term “efficacy sample” whereas “effectiveness” is used elsewhere (for e.g. in the study objectives and introduction). Please harmonize.
9. Figure 1: if I understand the figure correctly, in the box “Patients who did not use Papilocare® at least once (n=50)” on the left side of the figure, “excluded from the SAFETY analysis” should be added.
Results
1. Line 139: “Respecting sexual behavior pattern” à Did you mean “regarding sexual behavior pattern”?
2. Table 1:
- “Relevant disease/surgery” à perhaps it is worth specifying here that this does not include gynecological disease/surgery? (I assume)
- “Number of partners”: specify within the last month
- “Use of condoms”: is this also within the last month? (this is my understanding based on the footnote n°5)
- “HPV Vaccine” – “non-valuated” à what does this mean? I would assume “unknown” or “Not available”, but in the footnote n°6, you write that there are only 5 patients without information on HPV vaccination (vs 15 people in this category)
- Results from the biopsy: what does “HPV suggestive” refer to? In the data collection section in Methods, this category seems to be combined with “inflammatory” in one category “suggestive of inflammatory HPV”. Please clarify.
3. Table 2:
- why are some of the entries NA (for women >40 years)? It seems like these figures could be easily calculated.
- I would suggest harmonizing table headers to make them easier to read: e.g. either “visit 2”, “visit 3” & “visit 2 or 3” OR “6 months”, “12 months” & “6 or 12 months”
4. Figure 2: If I understand this figure correctly, in panel A for example, 30 patients (78-48) did not clear their HPV at 6 months but did not pursue the treatment. Is this because they had no more ASC-US/LSIL? Or lack of compliance? This should be explained in the results section.
5. Line 197: “basal biopsy” à did you mean “baseline biopsy”?
6. Line 198: “in which CIN-1 was the most frequent cytological result” à should be replaced by “histological result” (biopsies are analyzed by histology, not cytology). I would also suggest adding the proportion of CIN1 here.
7. Lines 199-200: “most of them did not show changes vs baseline” à please specify which proportion.
8. Lines 207-208: “In both visits, most of the participants approved the product” à it is not clear what is meant by “approved the product”. I would suggest rephrasing as “most of the participants gave a score of ≥5/10”
9. In general for the main results (ASC-US/LSIL regression and HPV clearance), it would be good to include 95% confidence intervals.
10. It would be interesting to provide the number/proportion of persistent ASC-US & LSIL at follow-up separately, as studies have shown different natural regression rates for these two types of lesions (e.g. Schlecht et al, https://doi.org/10.1093/jnci/djg037)
Discussion
1. Lines 218-220: “Results of this real-life study strengthen previously published data, in which Papilocare® emerges as an effective and safe option for the treatment of HPV-induced low-grade cervical lesions. HPV and low-grade lesions usually clear and regress spontaneously.” à add references for these 2 sentences.
2. Lines 223-225: “On the other hand, systematic reviews have established that the excisional and ablative treatment approach for cervical intraepithelial neoplasia is conflicting for specialists.” à add reference, and explain in what it is conflicting.
3. Line 228: replace monitorization by monitoring. Reference #20 is from the WHO, not Spanish guidelines, please change.
4. Lines 229-230: “Papilocare®, a Coriolus versicolor-based vaginal gel which has shown to improve the epithelialization of cervical mucosa and the composition of vaginal microbiota” à this should be in the introduction, not discussion.
5. Lines 236 & 240: please report p-values.
6. Lines 236-237: the overall primary result found in the current study (67% regression at 6 months) is actually closer to the control group in the randomized controlled trial (64.5%) than the treated group (84.9%). This contradicts the article’s conclusion that the treatment appears effective in a real-life setting.
Further, as there is no control group, the study results should be compared to previous findings on the spontaneous regression of ASC-US/LSIL lesions (e.g. Schlecht et al: https://doi.org/10.1093/jnci/djg037) and on natural HPV clearance in women (e.g. Rodriguez et al: https://doi.org/10.1093/jnci/djq001 , Winer et al: https://doi.org/10.1158/1055-9965.EPI-10-1108), which do not appear very different from the results after use of the Papilocare gel in the general population. However, there does appear to be a difference in the >40 y.o. population (in favor of the gel) that would warrant further research in a study comprising a control group.
6. Line 258: I wouldn’t consider a sample of <200 participants a “high number of patients”.
7. Lines 270-271: “a definitive conclusion cannot be made due to the low levels of biopsies analysed” à definitive conclusion on what? Please clarify.
8. Reference 23 is not valid, the link provided leads to a server error.
9. Further stratification of HPV type (e.g. HPV 16/18) would have been interesting to analyze as part of the HPV clearance outcome, as the rate of spontaneous regression and risk of invasive disease vary according to HPV type.
10. There is no mention on the level of compliance of participants having received the treatment (how many took the treatment as prescribed, how many stopped sooner than recommended) which appears crucial given the considerable length/frequency of treatment recommended. This should be discussed.
Conclusion
Lines 274-275: “The vaginal gel Papilocare® has shown to repair HPV-dependent low-grade cervical lesions and lead to successful HPV clearance in routine clinical practice” à this conclusion is misleading. The effectiveness of a therapeutical product cannot be demonstrated through an observational study with no control arm. This choice of study design is appropriate for descriptive purposes and to study acceptability/tolerability in real life, but does not allow to draw any conclusions on the effectiveness of the product.
The level of English is very good, only minor mistakes noticed.
Author Response
Dear Editor, Cancers
Enclosed please find the revised version of the manuscript, REAL-LIFE EFFECTIVENESS OF A MULTI-INGREDIENT CORIOLUS VERSICOLOR-BASED VAGINAL GEL IN WOMEN WITH HPV–DEPENDENT CERVICAL LESIONS: THE PAPILOBS PROSPECTIVE STUDY, for your consideration for publication in Cancers.
In the following pages, we have addressed the two reviewers’ comments point by point. We hope this revised version of the manuscript is now suitable for publication.
Thank you in advance for considering this manuscript. We look forward to hearing from you.
Cordially,
Javier Cortés Bordoy, MD, PhD.
Reviewer 2:
General concept comments
Overall, the paper is well structured, the introduction provides a strong background and justification of the study, and the results are clearly presented. My main comments below are related to the methodology and interpretation of results.
The main issue in this paper is the choice of study design, specifically the absence of a control arm, to answer the objectives regarding effectiveness of the therapeutic product tested (repair of low-grade cervical lesions and HPV clearance). In its current form, the study design is only adequate to inform about acceptability/safety outcomes (taking into consideration however the relatively small sample size) and descriptive results of cytological/HPV infection outcomes.
ANSWER: Since the design of this study does not includes a control arm, as we agree with the fact that efficacy is measured in randomized controlled trials (RCT), we have changed the term “effectiveness” for “effect” to be consistent with what is more in agreement with a descriptive study.
In this single-arm observational study we describe the effect in a given cohort representing the general population in real life conditions, which could potentially deviate significantly from the previously reported data in the PALOMA RCT. Some of these relevant differences may come from the population, which is usually more heterogeneous in the observational studies, the status of the pathology, how the product is applied, the adherence or even how the product is stored. PAPILOBS gives valuable information to the practitioners about how the product might improve the spontaneous regression rates and viral clearance, giving the chance of comparing this cohort with their experience and generally accepted and published spontaneous resolution rates.
Considering the previous point, the current conclusion is misleading as there is no way of knowing whether regression of ASCUS/LSIL and HPV clearance would have been lower without any treatment provided (which is currently the standard approach to low-grade lesions). In fact, previously published papers show similar results as presented in this paper for the natural regression of low-grade lesions (Schlecht et al: https://doi.org/10.1093/jnci/djg037) as well as for HPV clearance in women aged <40 years (Rodriguez et al: https://doi.org/10.1093/jnci/djq001 , Winer et al: https://doi.org/10.1158/1055-9965.EPI-10-1108).
ANSWER: Although we highly value the bibliography provided by the reviewer, and we agree with the conclusions made in these publications, two of the publications (Schlecht et al., 2021 and Winer et al., 2003), are difficult to compare with the PAPILOBS study as the average age of the cohorts of both studies is 18-19-year-old. As we know, age deeply impacts both the regression and clearance rates. When considering age, spontaneous regression rates of CIN lesions diminish from 44.7% in women under 25 to 24.9% in patients over 40[1]. In this regard, screening guidelines (including the Spanish and USA guidelines) do not recommend including patients ≤ 25-year-old as the regression rates are very high in these patients[2],[3].
However, the cohort presented in PAPILOBS is a 38.7-year-old cohort that still has regression and clearance rates that are much higher than the ones that we would expect for a cohort of this age, showing promising results.
The regression rates are widely described in the literature. The study of 570 women with LSIL cytology with a mean age of 36.0 years (similar cohort to Papilobs that was 38.7-year-old on average) showed that the cumulative probability of regression in the 2-year follow-up was 62.3%[4]. Furthermore, it is estimated that 35-40% normalise CIN-1 lesions within a year and 40-60% in two years[5],[6],[7]. In contrast, the normalisation rates found in PAPILOBS are 67% at 6 months and 77.1% at 12 months.
- The paper provides no information on the participants’ compliance to treatment, which would have been of interest considering the long period (6 to 12 months) and frequent applications (every day or on alternate days) of an intravaginal product. The study design would further be more appropriate to observe compliance than to assess effectiveness.
ANSWER: The protocol was designed to evaluate the repair of the cervical low-grade lesions measured through the percentage of patients with normalized cervical cytology together with concordant colposcopy images. We agree with the reviewer on the interest of these studies in reporting compliance and satisfaction of the patient with the treatment[8], which were also assessed in this study. On the other hand, HPV clearance, the level of satisfaction with the product and the tolerability of Papilocare® were also assessed. Therefore, we have added the following information to the manuscript:
Material and methods: To measure the level of satisfaction visual analogue scale (EVA) that comprised values from 0 (not satisfied) to 10 (very satisfied) was used. The patient self-assessed the satisfaction level during each study visit. Adherence to the prescribed treatment was assessed by direct question to patient in each study visit (lines 136-140).
Results: Satisfaction with treatment remained between 7.5 and 8.0 in both visits (7.87 at 6 months and 7,51 and at 12 months). Adherence to the prescribed treatment was assessed by direct question, which was very high after 6 months treatment (visit 2) since 94.2% of patients were compliant with the treatment. Remarkably, after 12 months treatment (visit 3) the compliance with the treatment remained very high, with an adherence rate of 98.4% (lines 259-262).
Discussion: Compared to the PALOMA study, adherence was similar after six months of treatment (94.2% vs 94.3%), as well as tolerability profile (three AEs vs 7 AEs considered related to the product under investigation). In the case of adherence, the results were high considering the long periods of product application (6 and 12 months) and frequent uses of the product under investigation (daily or on alternate days), which shows the awareness of patients towards this pathology (lines 324-329).
Conclusion: The vaginal gel Papilocare® has shown a beneficial effect in the repair of HPV-dependent low-grade cervical lesions and lead to successful HPV clearance in routine clinical practice. In addition, it has shown an adequate tolerability profile, with a high satisfaction and adherence rates among the women that used it. These results are in line with those observed in the PALOMA clinical trial and in other observational studies, which strengthens the role of this vaginal gel as a beneficial alternative to the simple watchful waiting approach in HPV-related low-grade cervical lesions (lines 348-354).
- The generalizability of results is difficult to assess as insufficient information is provided regarding study sites and the procedure for participant selection/recruitment.
ANSWER: Further information about participant selection and recruitment (lines 83-84, lines 88-92) has been included in the Material and Methods sections of the manuscript, while data regarding study sites has been provided in a new supplementary table (line 82).
Specific comments
Abstract
1. Line 18: I would suggest changing the sentence to “Infection may lead to different grade…”
ANSWER: This modification has been implemented as suggested (line 18).
Line 26: “by the end of the study” à please clarify what this means.
ANSWER: It has been changed to “at 6 or 12 months” as reported in Table 2 (lines 26-27).
- The conclusion of the abstract is not supported by the study design (no control group - no evidence that HPV clearance/cervical lesion repair would have been lower with no treatment) (see further comments in conclusion section)
ANSWER: The sentence “The vaginal gel Papilocare® has been shown to repair HPV-dependent low-grade cervical lesions, leading to a successful HPV clearance in routine clinical practice.” has been changed to “The vaginal gel Papilocare® has shown a beneficial effect in the repair HPV-dependent low-grade cervical and favourable tolerability in routine clinical practice.” (lines 30-31).
Introduction
1. The scientific background and rationale for the study are clearly explained.
References #9 & #10 (Hoffman et al. & Major et al.) do not support what is stated in the introduction at the place where they are cited (“The persistence of the virus is strongly linked to progression to more advanced precancerous lesions and, eventually to cancer [9,10].”)
ANSWER: The inclusion of reference 9 is supported on the following sentence: “Persistent high-risk human papillomavirus (HR-HPV) infection is strongly and consistently associated with high-grade cervical intraepithelial neoplasia (CIN) grade 2/3 acquisition and is considered essential for the progression of cervical precancer to invasive cervical cancer (ICC).”
The reference 10 has been changed to Burd, E.M. Human papillomavirus and cervical cancer. Clin. Microbiol. Rev. 2003, 16, 1–17, in which is stated that “. It is plausible that high-risk HPV infection occurs early in life, may persist, and, in association with other factors promoting cell transformation, may lead to a gradual progression to more severe disease.”
We believe that the sentence would be better reference by including the term HR HPV.
Lines 59-60: “Such procedure should […] avoid future over-treatment” à please explain here what is meant by “avoid future over-treatment” (this should be better argumented as some may consider treating low-grade cervical lesions as “overtreatment”).
ANSWER: This part of the sentence has been removed to avoid misunderstanding.
Line 63: “and other ingredients” à please detail or give a reference to the full description of the product used (ideally in the methods section).
ANSWER: The composition has been included in the manuscript (lines 66-70). “In this context, Papilocare® (Procare Health, Spain) is a vaginal gel including versicolor, that combines components with moisturizing, tissue regeneration and microbiota balancing properties (hyaluronic acid, Asian centella, Aloe vera and α-glucan oligosaccharide)[9] with ingredients that have positive effects on HPV-dependent cervical lesions and HPV clearance (Coriolus versicolor, Azadirachta indica and carboxymethyl-β-glucan)[10],[11].”
Study main objectives are clearly stated. However, I would also recommend stating the secondary objective of evaluation HPV clearance.
ANSWER: This information is already in the manuscript. “Secondary objectives included the HPV clearance, the level of satisfaction with the product and the tolerability of Papilocare” (lines 123-124).
Methods
1. The study type is well described. However, many other elements are missing in the methods section:
- What were the study sites?
ANSWER: These sites have been included in Supplementary Table 1, reporting different centres that participated in the study and recruited at least 1 patient.
- What was the recruitment procedure?
ANSWER: The following information has been added regarding inclusion/exclusion criteria: Another inclusion criteria included that the patients were prescribed Papilocare® based on medical decision prior to their participation in the study (lines 90-92).
- Data collection: how was cytology obtained? Who conducted the analyses? Same for biopsy analyses.
ANSWER: Since it is an observational study, both procedures were performed according to routinary clinical practice.
- How was satisfaction assessed (self-completed questionnaire vs filled by a HCP, on-line vs in person? Timing of the assessment)?
ANSWER: This information has been included: “To measure the level of satisfaction visual analogue scale (EVA) that comprised values from 0 (not satisfied) to 10 (very satisfied) was used. The patient self-assessed the satisfaction level during each visit to the center..” (lines 136-139).
- Evaluation of tolerability: how/when were side effects reported?
ANSWER: This information has been added to the manuscript: “Description, intensity and severity, relation with the treatment, management and resolution of adverse events were collected using the adverse events sheet of the Case Report Form (CRF). The information was collected by the Principal Investigator (PI) during each visit by direct question. Patients received a Patient Information Sheet with the contact details of the PI or some of his or collaborators for any incident potentially related to the study” (lines 140-145).
- Baseline characteristics: how were they obtained?
ANSWER: All data from the clinical study was collected in the CRF. Baseline data was gathered during the recruitment visit either by direct question to the patient or during the medical exploration (lines 140-143).
- Was follow-up part of the routine clinical care or only for the study? Describe exactly which tests were done at which visit (HPV, cytology, colposcopy done systematically at all 3 visits?).
ANSWER: Since this is an observational study, the follow-up is part of clinical practice. As previously stated, data obtained from 6 and 12-months was collected in the same way as baseline data. In all visits, information regarding normalization, colposcopy, citology and HPV clearance was gathered.
- Which HPV types were considered high-risk (even if already mentioned in the introduction, please confirm that the same 14 types were considered high-risk in the analysis).
ANSWER: The subtypes considered as HR are the following ones: 16, 18, 31, 33, 35, 39, 45, 51, 52, 56, 58, 59, and 68 (lines 135-136).
Line 75: #NCT04199260 à is this the clinicaltrial.gov number? Please specify. ANSWER: Yes, it is. It has been indicated (line 82).
Line 79: “with concordant colposcopy image” à please explain what is meant by this (how was it determined whether colposcopy was concordant with cytology, which criteria were used).
ANSWER: A colposcopy result was considered as concordant with cytological results of ASCUS, LSIL o AGUS for those colposcopic findings classified as normal or abnormal in grade 1 (minor) accordingly to the definition from the committee of the International Federation of Cervical Patology and Colposcopy (IFCPC), defined at Rio World Congress, 5th July 2011[12] (lines 132-135).
- Line 81: please specify what the contraindications are to the use of the Papilocare gel.
ANSWER: Papilocare should not be used in case of hypersensibility to any of its components, in case of itching or burning sensation, the application must stop and the patients should ask their doctor. This claim regarding hypersensibility has been included (lines 94-95).
Additionally, there is no clinical data about the use of Papilocare during pregnancy and- vaginal contraceptives, so in this case, the use of the product must be done under medical supervision.
Lines 92-97: this information should go in another section of the methods (e.g. study procedures), not data collection.
ANSWER: We agree, therefore, the information has been moved to study design section (lines 103-110).
Lines 94-95: Please clarify. Did all partcipants with either normal cytology/colposcopy OR HPV clearance leave the study at 6 months? For example if cyto/colpo was pathological but the HPV was cleared, were they not followed up until 12 months? This also contradicts the following sentence on lines 95-96.
ANSWER: Further information regarding this topic has been added to the manuscript (lines 106-107).
Line 105: which HPV PCR assay was used?
ANSWER: HPV was determined by the HPV diagnostic test according to the usual clinical practice in each site.
Line 116: you use here the term “efficacy sample” whereas “effectiveness” is used elsewhere (for e.g. in the study objectives and introduction). Please harmonize. ANSWER: Taking into account that this is a descriptive observational real-world evidence study, the term efficacy has been changed to effect throughout all the manuscript.
Figure 1: if I understand the figure correctly, in the box “Patients who did not use Papilocare® at least once (n=50)” on the left side of the figure, “excluded from the SAFETY analysis” should be added.
ANSWER: Figure 1 has been changed to reflect this change. In fact, these patients were also excluded from the effect data, since it was also not available.
Results
1. Line 139: “Respecting sexual behavior pattern” à Did you mean “regarding sexual behavior pattern”?
ANSWER: The sentence has been changed accordingly (line 183).
Table 1:
- “Relevant disease/surgery” à perhaps it is worth specifying here that this does not include gynecological disease/surgery?
ANSWER: Additional details have been added to Table 1 to reflect this information.
- “Number of partners”: specify within the last month
ANSWER: This specification has been added to Table 1.
- “Use of condoms”: is this also within the last month? (this is my understanding based on the footnote n°5)
ANSWER: This specification has been added to Table 1.
- “HPV Vaccine” – “non-valuated” à what does this mean? I would assume “unknown” or “Not available”, but in the footnote n°6, you write that there are only 5 patients without information on HPV vaccination (vs 15 people in this category).
ANSWER: Non-valuated has been changed to unknown, since no information was gathered from these patients about this topic.
- Results from the biopsy: what does “HPV suggestive” refer to? In the data collection section in Methods, this category seems to be combined with “inflammatory” in one category “suggestive of inflammatory HPV”. Please clarify please, unify to “suggestive of inflammatory HPV”
ANSWER: Both categories have been combined into the "suggestive of inflammatory HPV” variable.
Table 2:
- why are some of the entries NA (for women >40 years)? It seems like these figures could be easily calculated.
ANSWER: This data has been included to Table 2.
- I would suggest harmonizing table headers to make them easier to read: e.g. either “visit 2”, “visit 3” & “visit 2 or 3” OR “6 months”, “12 months” & “6 or 12 months”
ANSWER: Table heading have been harmonized according to the proposed format.
Figure 2: If I understand this figure correctly, in panel A for example, 30 patients (78-48) did not clear their HPV at 6 months but did not pursue the treatment. Is this because they had no more ASCUS/LSIL? Or lack of compliance? This should be explained in the results section.
ANSWER: We only have this data from the general sample (panel A of Figure 1). From the 30 patients, 18 did not enter due to either their own or their physician’s decision. Out from the 12 patients that entered, 5 were lost during the follow-up, 3 left due to their physician’s decision, 1 withdrew her consent and 3 their situation was unknown. This information has been added to the manuscript (lines 225-228).
Line 197: “basal biopsy” à did you mean “baseline biopsy”?
ANSWER: This modification has been implemented in the manuscript (line 244).
Line 198: “in which CIN-1 was the most frequent cytological result” à should be replaced by “histological result” (biopsies are analyzed by histology, not cytology). I would also suggest adding the proportion of CIN1 here.
ANSWER: The sentence has been rewritten as suggested (line 245).
Lines 199-200: “most of them did not show changes vs baseline” à please specify which proportion.
ANSWER: Further information has been added to the sentence (lines 247-248).
Lines 207-208: “In both visits, most of the participants approved the product” à it is not clear what is meant by “approved the product”. I would suggest rephrasing as “most of the participants gave a score of ≥5/10”
ANSWER: The sentence has been rewritten as suggested (lines 256-257).
- In general for the main results (ASC-US/LSIL regression and HPV clearance), it would be good to include 95% confidence intervals.
ANSWER: Values for regression and HPV clearance are represented as the number of patients that achieved either regression or clearance and their percentage, since it was only studied whether patients met criteria for achieving these two outcomes. Therefore, since they are represented as categorical variables, it does not make sense adding 95% confidence intervals, as it is only useful for quantitative variables.
It would be interesting to provide the number/proportion of persistent ASC-US & LSIL at follow-up separately, as studies have shown different natural regression rates for these two types of lesions (e.g. Schlecht et al, https://doi.org/10.1093/jnci/djg037)
ANSWER: The statistical report and design of the study did not include performing a follow-up depending on these two types of lesions, so we do not have this data available. However, it is an interesting topic to take into account for future research.
Discussion
Lines 218-220: “Results of this real-life study strengthen previously published data, in which Papilocare® emerges as an effective and safe option for the treatment of HPV-induced low-grade cervical lesions. HPV and low-grade lesions usually clear and regress spontaneously.” à add references for these 2 sentences.
ANSWER: References have been added to the sentence (lines 271-273).
Lines 223-225: “On the other hand, systematic reviews have established that the excisional and ablative treatment approach for cervical intraepithelial neoplasia is conflicting for specialists.” à add reference, and explain in what it is conflicting.
ANSWER: A reference and further explanation has been added to that sentence (lines 277-280).
- Line 228: replace monitorization by monitoring. Reference #20 is from the WHO, not Spanish guidelines, please change.
ANSWER: These changes have been applied to the manuscript.
Lines 229-230: “Papilocare®, a Coriolus versicolor-based vaginal gel which has shown to improve the epithelialization of cervical mucosa and the composition of vaginal microbiota” à this should be in the introduction, not discussion.
ANSWER: We think that this sentence adds context to the conclusion section of the manuscript.
Lines 236 & 240: please report p-values.
ANSWER: p-value has been added to the sentence (line 292).
Lines 236-237: the overall primary result found in the current study (67% regression at 6 months) is actually closer to the control group in the randomized controlled trial (64.5%) than the treated group (84.9%). This contradicts the article’s conclusion that the treatment appears effective in a real-life setting.
Further, as there is no control group, the study results should be compared to previous findings on the spontaneous regression of ASC-US/LSIL lesions (e.g. Schlecht et al: https://doi.org/10.1093/jnci/djg037) and on natural HPV clearance in women (e.g. Rodriguez et al: https://doi.org/10.1093/jnci/djq001 , Winer et al: https://doi.org/10.1158/1055-9965.EPI-10-1108), which do not appear very different from the results after use of the Papilocare gel in the general population. However, there does appear to be a difference in the >40 y.o. population (in favor of the gel) that would warrant further research in a study comprising a control group.
ANSWER: The PALOMA and PAPILOBS study populations and situations are not directly comparable. The data from the studies provided by the reviewer also cannot be compared with those of PAPILOBS because the populations were different in age and, in the case of the studies provided, the populations regressed their lesions and cleared HPV more easily. In addition, the PAPILOBS data, compared to studies or literature data, show a higher degree of regression and clearance. Therefore, with this comparison, even without a control arm in this study, it can be concluded that Papilocare has a beneficial effect on regression of low-grade cervical lesions and HPV clearance, and that these results are aligned with those of PALOMA and other observational studies.. Regarding the analysis of women over 40 years old, it is a very interesting research line for further studies that we are going to take into account.
Line 258: I wouldn’t consider a sample of <200 participants a “high number of patients”.
ANSWER: The sentence has been modified (lines 330-331).
- Lines 270-271: “a definitive conclusion cannot be made due to the low levels of biopsies analysed” à definitive conclusion on what? Please clarify.
ANSWER: Further clarification has been added to the sentence (lines 344-345).
Reference 23 is not valid, the link provided leads to a server error.
ANSWER: The link has been modified with the correct one (lines 483-485).
Further stratification of HPV type (e.g. HPV 16/18) would have been interesting to analyze as part of the HPV clearance outcome, as the rate of spontaneous regression and risk of invasive disease vary according to HPV type.
ANSWER: The design of the study did not include performing an analysis regarding the HPV type, so we do not have this data available. However, it is an interesting topic to consider for future studies.
There is no mention on the level of compliance of participants having received the treatment (how many took the treatment as prescribed, how many stopped sooner than recommended) which appears crucial given the considerable length/frequency of treatment recommended. This should be discussed.
ANSWER: Adherence to the prescribed treatment was assessed by direct question. We found that the adherence to the treatment was very high, after 6 months treatment (visit 2) 94.2% were compliant with the treatment. Remarkably, after 12 months treatment (visit 3) the compliance with the treatment remained very high, being a 98.4% (lines 139-140, lines 259-262).
Conclusion
Lines 274-275: “The vaginal gel Papilocare® has shown to repair HPV-dependent low-grade cervical lesions and lead to successful HPV clearance in routine clinical practice” à this conclusion is misleading. The effectiveness of a therapeutical product cannot be demonstrated through an observational study with no control arm. This choice of study design is appropriate for descriptive purposes and to study acceptability/tolerability in real life, but does not allow to draw any conclusions on the effectiveness of the product.
ANSWER: The conclusion section of the manuscript has been modified (lines 348-354).
Comments on the Quality of English Language
The level of English is very good, only minor mistakes noticed.
[1]Bekos C, et al. Influence of age on histologic outcome of cervical intraepithelial neoplasia during observational management: results from large cohort, systematic review, meta-analysis. Sci Rep. 2018;8(1):6383. doi: 10.1038/s41598-018-24882-2.
[2]AEPCC-Guía: PREVENCIÓN SECUNDARIA DEL CANCER DE CUELLO DEL ÚTERO, 2022. CONDUCTA CLÍNICA ANTE RESULTADOS ANORMALES DE LAS PRUEBAS DE CRIBADO. Coordinador: Torné A. Secretaria: del Pino M. Autores: Torné A; Andía, D; Bruni L; Centeno C; Coronado P; Cruz Quílez J; de la Fuente J; de Sanjosé S; Granados R; Ibáñez R; Lloveras B; Lubrano A Matías Guiu X; Medina N; Ordi J; Ramírez M; del Pino M.
[3]Perkins RB, et al. 2019 ASCCP Risk-Based Management Consensus Guidelines for Abnormal Cervical Cancer Screening Tests and Cancer Precursors. J Low Genit Tract Dis. 2020;24(2):102-131. doi: 10.1097/LGT.0000000000000525.
[4]Matsumoto K, et al. Predicting the progression of cervical precursor lesions by human papillomavirus genotyping: a prospective cohort study. Int J Cancer. 2011;128(12):2898-910. doi: 10.1002/ijc.25630.
[5]Dalstein V, et al. Persistence and load of high-risk HPV are predictors for development of high-grade cervical lesions: a longitudinal French cohort study. Int J Cancer. 2003;106(3):396-403. doi: 10.1002/ijc.11222.
[6]Bansal N, et al. Natural history of established low grade cervical intraepithelial (CIN 1) lesions. Anticancer Res. 2008;28(3B):1763-6.
[7]Melnikow J, et al. Natural history of cervical squamous intraepithelial lesions: a meta-analysis. Obstet Gynecol. 1998;92(4 Pt 2):727-35. doi: 10.1016/s0029-7844(98)00245-2.
[8]Barbosa CD, et al. A literature review to explore the link between treatment satisfaction and adherence, compliance, and persistence. Patient Prefer Adherence. 2012;6:39-48. doi: 10.2147/PPA.S24752.
[9]Lee JH, et al. Asiaticoside enhances normal human skin cell migration, attachment and growth in vitro wound healing model. Phytomedicine. 2012 O;19(13):1223-7. doi: 10.1016/j.phymed.2012.08.002.
[10]Shukla S, et al. Elimination of high-risk human papillomavirus type HPV16 infection by 'Praneem' polyherbal tablet in women with early cervical intraepithelial lesions. J Cancer Res Clin Oncol. 2009;135(12):1701-9. doi: 10.1007/s00432-009-0617-1.
[11]Scardamaglia P, et al. Efficacia del carbossimetilbetaglucano nella regressione delle alterazioni citologiche cervicali di basso grado HPV correlate [Effectiveness of the treatment with beta-glucan in the HPV-CIN 1 lesions]. Minerva Ginecol. 2010;62(5):389-93.
[12]Bornstein J, et al. 2011 colposcopic terminology of the International Federation for Cervical Pathology and Colposcopy. Obstet Gynecol. 2012;120(1):166-72. doi: 10.1097/AOG.0b013e318254f90c.

Round 2
Reviewer 1 Report
no further comments
Author Response
The authors would like to thank the reviewer for the stimulating comments that help us to improve the quality of our manuscript. We have really appreciated the annotations.
Reviewer 2 Report
Thank you for the revised version of this paper. Below are my remaining comments in response to the revision. My main concern remains about the use of the term "effect" which is not appropriate for the selected study design.
General comment:
The authors changed the term “effectiveness” to “effect”, which is not satisfying as it still misleadingly suggests the observed regression of low-grade lesions is an effect of the vaginal treatment and not due to spontaneous regression (which can not be deduced from this study based on the chosen design). Please remove all reference to the “effect” of the gel or “effectiveness” when discussing results of this study. As already stated in the first review report, the observational study design used here is not appropriate to measure effect of the intervention.
Methods
- The new supplementary table with included study sites has not been provided.
- Lines 142-143: please confirm that the occurrence of adverse events was “collected by the principal investigator during each visit by direct question” (it seems unlikely that the PI would be present at each visit at different participating health facilities to ask this question)
Results
Table 1:
- the “not” added in “specifications of relevant not gynecological disease/surgery” seems to be a mistake.
- Footnote 6: n=5 patients without information about HPV vaccination contradicts n=15 with unknown HPV vaccination in the table. Please correct.
Table 2:
- The NAs have been replaced in the subsection “repair of lesions” but not “cytology result + coloposcopy” although it seems this information could be easily calculated from the data available for visits 2 and 3.
- Lines 225-229: these patients who have stopped taking the treatment despite still having persistent HPV infection should be counted as “non-adherent” which does not seem to be the case in the numbers reported in the results on adherence (line 262).
- The authors have answered that 95% confidence intervals are not appropriate for categorical variables, which is not correct. Please see for example Lancet. 2004 Nov 6-12;364(9446):1678-83. doi: 10.1016/S0140-6736(04)17354-6 (or any other paper reporting population proportions with their 95% confidence intervals, as is done in standard practice). This is also recommended in the STROBE reporting guidelines for observational studies (https://www.strobe-statement.org/).
Discussion
- The authors state that the PALOMA and PAPILOBS study populations are not comparable and therefore the similar regression rates in the treated group by papilocare in PAPILOBS (67%) as in the control (untreated) group in the randomized controlled trial PALOMA (64.5%) should not be compared. If that is the case, it should be explained in the paper why they consider that these 2 populations are not comparable and what makes the other studies selected to suggest a “positive effect” of the treatment (i.e. Matsumoto et al, Dalstein et al, Bansal et al, Melnikow et al) more comparable to the PAPILOBS study population… If this argument cannot be made, then I would suggest giving a more balanced interpretation of the results.
- Line 273: references 22 and 23 are unrelated to the sentence in which they are cited.
- Line 280: please change to “and some ablative treatments are linked to higher failure rates” (cold coagulation has comparable failure rates as LLETZ according to the cited reference).
- Line 282: an FAQ webpage is not an appropriate reference for a scientific paper. Please provide a valid reference from a scientific journal.
- Line 285: typo? (“elderlvaginal”)
- References 26 and 32: the links to the web pages do not work
Conclusion
It remains wrong to state that the study “has shown a beneficial effect” of the vaginal gel. Please change the wording. The same applies to the conclusion of the abstract.
I would suggest sticking to the facts by writing something like “we observed xx% regression of LSIL in women treated by papilocare, which is higher than spontaneous regression rates in previously published literature” or “the regression of LSIL in women treated by papilocare appeared to be higher than spontaneous regression rates”.
The text added in the revisions contains many English mistakes and should be revised.
Author Response
Dear Editor, Cancers
Enclosed please find the revised version of the manuscript, REAL-LIFE EFFECTIVENESS OF A MULTI-INGREDIENT CORIOLUS VERSICOLOR-BASED VAGINAL GEL IN WOMEN WITH HPV–DEPENDENT CERVICAL LESIONS: THE PAPILOBS PROSPECTIVE STUDY, for your consideration for publication in Cancers.
In the following pages, we have addressed the reviewers’ comments point by point. We hope this revised version of the manuscript is now suitable for publication.
Thank you in advance for considering this manuscript. We look forward to hearing from you.
Cordially,
Javier Cortés Bordoy, MD, PhD.
General comment: The authors changed the term “effectiveness” to “effect”, which is not satisfying as it still misleadingly suggests the observed regression of low-grade lesions is an effect of the vaginal treatment and not due to spontaneous regression (which can not be deduced from this study based on the chosen design). Please remove all reference to the “effect” of the gel or “effectiveness” when discussing results of this study. As already stated in the first review report, the observational study design used here is not appropriate to measure effect of the intervention.
Response: We have removed all reference to the “effect” or “effectiveness” of the gel from the study objective and when discussing the results of this study. However, “efficacy” or “effect” or “effectiveness terms” have not been changed when discussing the results of other studies in which these words were used.
Reviewer #2
Comment: Methods: The new supplementary table with included study sites has not been provided.
Response: The Supplementary Material has been updated with a new version including the new table.
Comment: Lines 142-143: please confirm that the occurrence of adverse events was “collected by the principal investigator during each visit by direct question” (it seems unlikely that the PI would be present at each visit at different participating health facilities to ask this question)
Response: In this study there was one principal investigator per center, with data from each one being collected either by the principal investigators and/or collaborators. It has been added to the manuscript (lines 136).
Comment: Results: Table 1: the “not” added in “specifications of relevant not gynecological disease/surgery” seems to be a mistake.
Response: The “not” has been removed from the table caption.
Comment: Footnote 6: n=5 patients without information about HPV vaccination contradicts n=15 with unknown HPV vaccination in the table. Please correct.
Response: Table 1 has been modified and the footnote removed (line 185-187).
Comment: Table 2: The NAs have been replaced in the subsection “repair of lesions” but not “cytology result + coloposcopy” although it seems this information could be easily calculated from the data available for visits 2 and 3.
Response: We have added the missing results of “cytology result + colposcopy” to Table 2.
Comment: Lines 225-229: these patients who have stopped taking the treatment despite still having persistent HPV infection should be counted as “non-adherent” which does not seem to be the case in the numbers reported in the results on adherence (line 262).
Response: The percentage included in that specific sentence does not take into account neither the patients who did not continue in the study after visit 2 by their own or their physician’s decision, even though they had not cleared HPV (n=18), nor those who were lost/dropped out between visits 2 and 3 (n=12). Therefore, the percentage of adherence included only those patients who completed the study at visit 3. As stated in the modifications performed in the previous round, data was not available from 30 patients (18 did not continue treatment due to either, their own or their physician’s decision, and out from the 12 patients that continued, 5 were lost during the follow-up, 3 left due to their physician’s decision, 1 withdrew her consent and 3 from which their situation was unknown). Those 18 patients cannot be considered in the poor adherence group because they were not prescribed with the treatment from visit 2. The remaining 12 who lost/dropped out the study due to above mentioned reasons might be considered as having poor adherence to treatment. An additional sentence including this last statement has been added to the manuscript (lines 218-226).
Comment: The authors have answered that 95% confidence intervals are not appropriate for categorical variables, which is not correct. Please see for example Lancet. 2004 Nov 6-12;364(9446):1678-83. doi: 10.1016/S0140-6736(04)17354-6 (or any other paper reporting population proportions with their 95% confidence intervals, as is done in standard practice). This is also recommended in the STROBE reporting guidelines for observational studies (https://www.strobe-statement.org/).
Response: The reviewer was right and it was a misunderstanding on our part. 95% confidence intervals have been added to the text since Table 2 is already large and in order to not confuse the reader. These intervals have also been added to Figure 2 (lines 191-201 and lines 215-235).
Comment: Discussion: The authors state that the PALOMA and PAPILOBS study populations are not comparable and therefore the similar regression rates in the treated group by papilocare in PAPILOBS (67%) as in the control (untreated) group in the randomized controlled trial PALOMA (64.5%) should not be compared. If that is the case, it should be explained in the paper why they consider that these 2 populations are not comparable and what makes the other studies selected to suggest a “positive effect” of the treatment (i.e. Matsumoto et al, Dalstein et al, Bansal et al, Melnikow et al) more comparable to the PAPILOBS study population… If this argument cannot be made, then I would suggest giving a more balanced interpretation of the results.
Response: We have added an explanation about why these two studies cannot be compared. In addition, we have removed comparisons with the PALOMA study in the discussion regarding lesion’s regression and HPV clearance. We have also added the design of the studies from which comparisons are made in the discussion (lines 300-310).
Comment: Line 273: references 22 and 23 are unrelated to the sentence in which they are cited.
Response: These references have been removed from the sentence (line 278).
Comment: Line 280: please change to “and some ablative treatments are linked to higher failure rates” (cold coagulation has comparable failure rates as LLETZ according to the cited reference).
Response: It has been added to the manuscript as suggested (lines 284-285).
Comment: Line 282: an FAQ webpage is not an appropriate reference for a scientific paper. Please provide a valid reference from a scientific journal.
Response: A new reference has been provided (lines 492-492).
Comment: Line 285: typo? (“elderlvaginal”)
Response: The typo has been edited (line 290).
Comment: References 26 and 32: the links to the web pages do not work.
Response: We can access the link of reference 26 without problems. The link of reference 32, which linked the abstract book of the congress, does not seem to work anymore as the webpage from the congress does not provide access to it. Therefore, the link has been removed from that reference.
Comment: Conclusion: It remains wrong to state that the study “has shown a beneficial effect” of the vaginal gel. Please change the wording. The same applies to the conclusion of the abstract. I would suggest sticking to the facts by writing something like “we observed xx% regression of LSIL in women treated by papilocare, which is higher than spontaneous regression rates in previously published literature” or “the regression of LSIL in women treated by papilocare appeared to be higher than spontaneous regression rates”.
Response: The sentence has been changed in both the main text and the abstract as suggested (lines 33-35, lines 367-371).
